# High-Intensity Interval Training (HIIT) in Hypoxia Improves Maximal Aerobic Capacity More Than HIIT in Normoxia: A Systematic Review, Meta-Analysis, and Meta-Regression

**DOI:** 10.3390/ijerph192114261

**Published:** 2022-11-01

**Authors:** Ailsa Westmacott, Nilihan E. M. Sanal-Hayes, Marie McLaughlin, Jacqueline L. Mair, Lawrence D. Hayes

**Affiliations:** 1Sport and Physical Activity Research Institute, University of the West of Scotland, Glasgow G72 0LH, UK; 2Future Health Technologies, Singapore-ETH Centre, Campus for Research Excellence and Technological Enterprise, Singapore 138602, Singapore

**Keywords:** altitude, sprint, training, endurance, VO_2max_

## Abstract

The present study aimed to determine the effect of high intensity interval training (HIIT) in hypoxia on maximal oxygen uptake (VO_2max_) compared with HIIT in normoxia with a Preferred Reporting Items for Systematic Reviews and Meta-Analysis (PRISMA)-accordant meta-analysis and meta-regression. Studies which measured VO_2max_ following a minimum of 2 weeks intervention featuring HIIT in hypoxia versus HIIT in normoxia were included. From 119 originally identified titles, nine studies were included (*n* = 194 participants). Meta-analysis was conducted on change in (∆) VO_2max_ using standardised mean difference (SMD) and a random effects model. Meta-regression examined the relationship between the extent of environmental hypoxia (fractional inspired oxygen [FiO_2_]) and ∆VO_2max_ and intervention duration and ∆VO_2max_. The overall SMD for ∆VO_2max_ following HIIT in hypoxia was 1.14 (95% CI = 0.56–1.72; *p* < 0.001). Meta-regressions identified no significant relationship between FiO_2_ (coefficient estimate = 0.074, *p* = 0.852) or intervention duration (coefficient estimate = 0.071, *p* = 0.423) and ∆VO_2max_. In conclusion, HIIT in hypoxia improved VO_2max_ compared to HIIT in normoxia. Neither extent of hypoxia, nor training duration modified this effect, however the range in FiO_2_ was small, which limits interpretation of this meta-regression. Moreover, training duration is not the only training variable known to influence ∆VO_2max_, and does not appropriately capture total training stress or load. This meta-analysis provides pooled evidence that HIIT in hypoxia may be more efficacious at improving VO_2max_ than HIIT in normoxia. The application of these data suggest adding a hypoxic stimuli to a period of HIIT may be more effective at improving VO_2max_ than HIIT alone. Therefore, coaches and athletes with access to altitude (either natural or simulated) should consider implementing HIIT in hypoxia, rather than HIIT in normoxia where possible, assuming no negative side effects.

## 1. Introduction

### 1.1. Rationale

A combination of reduced barometric pressure (PB), or a reduced effective inspired fraction of oxygen (FiO_2_), leads to reduced inspired partial pressure of oxygen (PaO_2_) which ultimately results in the physiological state of hypoxia [1,2]. In human research concerning hypoxia, hypoxia can be examined through two means, firstly hypobaric hypoxia (PB < 760 mmHg; FiO_2_ = 20.9%) which generally reflects the state found on earth at altitude, and normobaric hypoxia (PB = 760 mmHg; FiO_2_ < 20%) which can be considered simulated altitude [3]. Chronic exposure to natural altitude stimulates renal production of erythropoietin (EPO), driving increased haemoglobin mass (Hb mass) and red blood cell (RBC) count [4,5]. This increases oxygen carrying capacity of the blood, with well reported associations between increased RBC and Hb mass and improved maximal oxygen uptake (VO_2max_) [6,7]. Therefore, athletes have been recommended to spend prolonged periods of time at moderate (2000–3000 m), to high altitude (>3000 m), to stimulate erythropoiesis [8]. Thus, ‘altitude training camps’ are widely used by professional and recreational athletes alike [9], and the early 1990s saw the popularisation of the ‘live-high, train-low’ (LHTL) paradigm by Levine and Stray- Gundersen [6]. In a pooled analysis of six previous experiments, studies adopting the LHTL and LHTH paradigms have shown improved test performance at sea-level, with a ~3% increase in VO_2max_ following altitude training compared to control (i.e., normoxic) participants [10]. More recently, hypoxic training methods have been further developed into live high train high (LHTH), intermittent hypoxic exposure at rest, and live low train high (LLTH) [11,12,13].

In the last decade, high-intensity interval training (HIIT) has become in vogue, evidenced by the American College of Sports Medicine (ACSM) reporting it as its number one fitness trend in 2014, and number two in 2020 [14]. HIIT can be defined as repeated bouts of high intensity effort followed by varied recovery times. Sprint-interval training (SIT; an ‘all-out’ derivative of HIIT) may also be considered under the HIIT umbrella. HIIT is typically around or below 100% VO_2max_ whilst SIT is all-out, an intensity above VO_2max_ [15,16,17,18,19,20,21,22]. Both HIIT and SIT are reportedly efficacious in improving VO_2max_ at sea level [19,21,22,23,24,25,26]. The understanding that exercise intensity is a potent stimuli for improving VO_2max_ is not a new discovery [27,28,29,30]. To these ends, HIIT is generally considered a more potent stimulus than moderate-intensity continuous training (MICT) for improving VO_2max_ (especially over a short intervention duration) [23,25,31,32,33]. In this context, a recent meta-analysis by Su and colleagues [25] indicated that HIIT in normoxia increased VO_2max_ more than MICT in normoxia, in overweight/obese individuals (aged 18–48 years), specifically when intervals were >2 min in duration, with a standard mean difference of 0.444 (95% confidence intervals [CI] = 0.037–0.851; Small magnitude of effect). Moreover, a seminal RCT [33] observed greater improvement in VO_2max_ following 4 × 4 min running with 3 min rests (Cohen’s *d* = 0.66) and 47 × 15 s of running with 15 s rests (Cohen’s *d* = 0.79) at 90–95% heart rate maximum compared to lactate threshold (Cohen’s *d* = 0.16) and sub-threshold training (Cohen’s *d* = 0.13).

Recently, HIIT training has been combined with hypoxia training with the aim of eliciting optimal training adaptations. LLTH methods allow athletes to continue to live at normoxia, whilst exposed to acute periods of hypoxia during training. Within LLTH there are different training methodologies, including continuous hypoxic training (CPT), interval hypoxic training (IHT), and repeated sprint training in hypoxia (RST). Several original investigations have now been conducted examining either HIIT or RST in hypoxia [12,34,35,36]. Gatterer et al. [37] published results from a pilot study noting HIIT and RST in hypoxia improved sea-level performance of the repeated sprint ability (RSA) and a Yo-Yo intermittent recovery test 2 (YYIR2), in addition to muscle re-oxygenation. Studies investigating physiological adaptations during HIIT in hypoxia have reported that different training methods in hypoxia elicit different training effects, including increased oxidative capacity (CPT), buffering capacity (IHT), and compensatory fiber-selective vasodilation (RST), respectively [12]. Overall, LLTH methods have been shown to stimulate non-haematological peripheral adaptations, such as muscular adaptations which promote energy metabolism, alongside increased perfusion, improving O_2_ utilisation and delivery, favouring sporting performance [11].

Despite previous reports suggesting training in hypoxia can augment HIIT-induced adaptations in VO_2max_ [37], a systematic review by Hopeller et al. [38] reported hypoxia supplementary to exercise training was not consistently advantageous for performance at sea level. Hamlin et al.’s [39] meta-analysis of HIIT-hypoxia training focusses specifically on populations participating in team sports, and on high intensity running (Yo-Yo IRT) after a hypoxic interventions. Therefore, there is a need for a meta-analysis with wider inclusion criteria (i.e., not just team sports players) with VO_2max_ as the primary outcome variable, as VO_2max_ is the gold standard of cardiorespiratory fitness measurement. Therefore, for a more coherent interpretation of the effects of HIIT in hypoxia vs. HIIT in normoxia, a quantitative pooled analysis of previous studies was necessary.

### 1.2. Objectives

Despite the abundance of studies investigating and reviewing hypoxia and HIIT separately, there was a lack of literature focusing on the effects of HIIT in hypoxia on VO_2max_. Therefore, the aim of this investigation was to conduct a meta-analysis on the effect of HIIT in hypoxia compared to HIIT in normoxia on VO_2max_. A secondary aim was to investigate study characteristics (i.e., degree of hypoxia, study duration) on magnitude of effect through meta-regressions.

## 2. Materials and Methods

### 2.1. Eligibility Criteria

This meta-analysis was conducted according to the Preferred Reporting Items for Systematic Reviews and Meta-Analysis (PRISMA) guidelines. Studies which met the following criteria were included: (1) full text manuscript; (2) not a review; (3) studies were required to have a control group within normoxic/sea-level environment or include pre-exercise intervention measures; (4) healthy participants of any sex aged 16–65 years; (5) studies were required to employ a HIIT intervention/programme for a minimum of 14 days. Furthermore, studies were required to have reported descriptive data, such as mean, standard deviation (SD), and sample size (*n*). If required, requests for details and full papers were submitted to the author(s). The primary aim was to investigate whether VO_2max_ was affected by HIIT in hypoxia (environmental or simulated). Therefore, only studies which directly measured (i.e., not estimated) VO_2max_ (ml·kg·min^1^ or L·min^−1^ pre- and post-intervention) were included. Within this review both randomised control trials (RCTs) and non-randomised control trials (CTs) were considered. Thus, studies without a control group (i.e., uncontrolled trials with a pre- to post-exposure design) were excluded from analysis.

### 2.2. Information Sources

PubMed, ScienceDirect, and SPORTDiscus were searched with no start date up until 10 February 2021. The search was performed within all fields and terms were “HIIT” AND “hypoxia”, “HIIT” AND “hypoxic”, “HIIT” AND “altitude”, “high-intensity interval training” AND “hypoxia”, “high-intensity interval training” AND “hypoxic”, “high-intensity interval training” AND “altitude”, “Sprint interval training” AND “hypoxia”, “Sprint interval training” AND “hypoxic”, and “Sprint interval training” AND “altitude”.

### 2.3. Study Selection

Following searches, obtained manuscripts were downloaded into a single reference manager (Zotero, 2016, Zotero version 5.0.96.1). Prior to eligibility screening the papers were sorted into a single reference list, with duplicates removed. Title and abstracts for all papers were screened for eligibility by two authors (A.W. and L.D.H.) with those that did not meet inclusion criteria excluded. Any disagreement between both reviewers was discussed in a consensus meeting. Out of the remaining manuscripts, those which examined HIIT in hypoxia were collated. Full text manuscripts were screened in depth and compared against inclusion and exclusion criteria. Following full text eligibility screening authors extracted participant data sets (sample size, *n*; age, mean ± SD), exercise modality (cycling, running, swimming, etc.), intervention method (HIIT, SIT, multi-component training, and interval training), intervention duration, altitude conditions (FiO_2_, or height above sea level, e.g., 3000 m) and VO_2max_ analysis method (Douglas bag, breath by breath gas analysis). Furthermore, manuscripts were coded as RCTs or CTs (Figure 1). Subsequently, all remaining papers were assessed against the Physiotherapy Evidence Database (PEDro) scale. The PEDro scale objectively assesses methodological quality of each study [40].

### 2.4. Data Collection Process

Information was imported into a spreadsheet, which was specifically designed for meta-analyses (Jamovi version 2.3.0.0, MAJOR package, https://www.jamovi.org, 2 October 2022). Data from both hypoxic and control groups were extracted from manuscripts: Change in (Δ)VO_2max_ (ml·kg·min^−1^ or l·min^−1^), effective FiO_2_, intervention duration, and sample size (n). For clarity, in an RCT or CT, the mean and SD ΔVO_2max_ from pre- to post-training in the experimental group and control group, plus the *n* of each group is entered into the spreadsheet (six data items). Where the mean ΔVO_2max_ was not reported, we subtracted pre-training VO_2max_ from post-training VO_2max_. Where the SD ΔVO_2max_ was not reported, it was calculated thusly: $$\sigma_{change}=\sqrt{\left(\sigma_{1}^{2}+\sigma_{2}^{2}-\left(2\cdot corr\cdot \sigma_{1}\cdot \sigma_{2}\right)\right)}$$
whereby: *corr* = correlation coefficient, a value of which describes the relationship between baseline and final VO_2max_ measurements over time. We used the correlation coefficient from Lawler et al. [41] (0.94) which was the correlation coefficient of pre-and post-intervention VO_2max_ in a group of trained and untrained male athletes (*n* = 13). In cases of missing data, authors were contacted via email and asked to provide necessary information. If no response was received, means and SDs were estimated from figures using computer software (Image J, Towson, MD, USA, Imagej.net (accessed on 10 April 2021).

### 2.5. Data Items

Standardised mean differences (SMD) expressed the intervention effect within each study [42] using a restricted maximum-likelihood model estimate. For clarity, the mean change in (∆) VO_2max_ in the hypoxic group, the SD of ∆VO_2max_ in the hypoxic group, the *n* of the hypoxic group, the mean ∆VO_2max_ in the normoxic (i.e., control) group, the SD of ∆VO_2max_ in the normoxic group, and the *n* of the normoxic group were used to calculate SMD. All studies had a control group so no uncontrolled trials were analysed. The alpha level (*p*) describes the probability of a type I error, and 95% was used as the confidence interval (CI) level. Statistical heterogeneity was quantified using the *I*^2^ statistics. An *I*^2^ value greater than 50% is classified as moderate to high between study heterogeneity. Due to the included studies being considered heterogeneous (*I*^2^ = 63%) a random effects meta-analysis was conducted. Funnel plots and the trim and fill method [43] assessed publication bias. The trim and fill method determines the number of studies necessary to eradicate publication bias from the funnel plot.

## 3. Results

### 3.1. Study Selection

Combined results from the three database searches identified 441 articles (Figure 1). After duplicates were removed a total of 119 titles and abstracts were screened for eligibility using the inclusion and exclusion criteria. We attempted to retrieve 37 records, and 33 reports were successfully retrieved and assessed for eligibility. Of the 33 screened, 24 papers were excluded, leaving nine full text manuscripts included within the final quantitative synthesis.

### 3.2. Study Characteristics

On completion of data pooling, nine studies were included in the analysis: seven were RCTs and two were control trials (Table 1). Within the nine studies, a total of 194 participants (men = 139, women = 55) were included. Studies were 2–13 weeks in duration (Table 2), and included running, cycling, swimming, or multi-component training. The PEDro scale determined quality of studies, and results indicated a mean score of 4 ± 1.

### 3.3. Meta-Analysis

The overall SMD for HIIT in hypoxia was 1.14 (95% CI = 0.56–1.72; *p* < 0.001; Figure 2). Heterogeneity (*I*^2^ = 67%) justified the use of a random effects model. The Richardson et al. [50] study was weighted the most within the meta-analysis (13%), whereas Zebrowska et al. [52] carried the least weight (9%). Visual inspection of the funnel plot (Figure 3) suggest studies were spread across both sides of the pooled SMD (i.e., without asymmetry) indicating low publication bias. The Trim and Fill method confirmed the number of inputted studies to eliminate publication bias was one, although the overall number of studies was an uneven number. In the next paragraph we describe our sensitivity analysis where we removed one studies to achieve a value of zero (i.e., plot symmetry).

We subsequently performed sensitivity analysis by removing the two studies which fell outside of the funnel [50,51]. This procedure did not cause a qualitative effect (i.e., the direction of overall effect). This resulted in a SMD of 1.51 (95% CI = 1.06–1.96; *p* < 0.001). We subsequently performed sensitivity analysis by removing the two studies which were not RCTs [44,49]. This procedure did not cause a qualitative effect and the SMD was 1.11 (95% CI = 0.35–1.86; *p* = 0.004). Finally, we removed the study by Jung et al. [47] to achieve a Trim and Fill value of 0 and the SMD was 1.01 (95% CI = 0.42–1.60; *p* < 0.01). Therefore, we believe results from the initial meta-analysis are robust against the analysis decisions.

### 3.4. Meta-Regressions

A random effects meta-regression examined the effect of intervention duration on SMD which indicated no relationship (coefficient estimate = 0.071, 95% CI = −0.103–0.245, *p =* 0.423). A random effects meta-regression examined the effect of effective FiO_2_ on SMD indicated no relationship (coefficient estimate = 0.074, 95% CI = −0.702–0.849, *p =* 0.852).

## 4. Discussion

### 4.1. Overview

The primary aim of this meta-analysis was to test whether VO_2max_ was affected by HIIT in hypoxia (environmental or simulated) more than HIIT in normoxia. The main findings were threefold. Firstly, HIIT in hypoxia increased VO_2max_ more than HIIT in normoxia. Secondly, meta-regression analysis suggested no relationship between intervention duration in weeks and SMD. Lastly, meta-regression analysis similarly suggested there was no relationship between effective FiO_2_ and SMD. Given that HIIT has undergone a recent surge in the literature, this meta-analysis provides pooled evidence that HIIT in hypoxia may be more effective in improving VO_2max_ than HIIT in normoxia.

### 4.2. The Effect of HIIT in Hypoxia on VO_2max_

When the studies were pooled, HIIT in hypoxia displayed a positive effect on VO_2max_, compared to normoxia, in which eight of nine studies demonstrated a positive SMD. However, the observed negative SMD in one study [51], was resultant of the normoxic group improving VO_2max_ (+5.6%), to a greater extent than the hypoxic group (+3.8%). Jung et al. [47] demonstrated the largest effect size of all included studies (SMD = 2.20). The result may not initially be surprising to the reader as participants in the Jung et al. [47] study exercising at 90–95% HR_max_, with a high training volume which included ten 5 min intervals per session (90 min, 3 d∙wk^−1^ for 6 weeks). Therefore, this large effect may be a repercussion of time spent ≥90% HR_max_ or VO_2max_ (they are highly related), as time spent at this intensity is known to determine training adaptations [53,54,55]. However, in a meta-analytical approach, the ∆VO_2max_ from hypoxic group would be compared to the ∆VO_2max_ in the normoxic group. As both groups underwent the same training regime, it is difficult to associate large SMDs to the training alone, as the ‘control’ normoxic group underwent analogous training to the ‘treatment’ hypoxic group.

Upon initial examination of the HIIT interventions, it was surprising that the study of Czuba et al. [45] exhibited a similar effect size (SMD = 2.13) to Jung and colleagues [47], as intervention duration was half that (60–75 min, 3 d∙wk^−1^ for 3 weeks) of Jung et al. [47]. Upon further inspection, although the main aspect of the interval training was 60–75 min, athletes actually spent 2 h per session (6 h wk^−1^) in hypoxia due to additional warm-ups, cool-downs, and general endurance riding. Therefore, this extra time in hypoxia at lower intensities may have provided an additional stimulus to hypoxic HIIT. Mechanisms behind hypoxia-induced improvements in VO_2max_ compared to volume- and intensity-matched normoxic training is not fully understood with regard to HIIT. However, exposure to acute hypoxia significantly reduces VO_2max_ [56]. It may be that as VO_2max_ acutely decreases with increasing altitude [56], the additional challenge of hypoxia increases the relative intensity of exercise, providing an added stimulus for adaptation. VO_2max_ declines to a larger extent in acute normobaric hypoxia compared to hypobaric hypoxia, occurring alongside a higher VEmax in hypobaric hypoxia [57]. Therefore, this may explain the differences in training adaptations between simulated [45,46,48,50,51,52] and natural altitude [44,49] studies. As time spent ≥90% HR_max_ or VO_2max_ determines training adaptations to HIIT [53,54,55], the addition of hypoxia to training may have increased time over this threshold intensity and thus aerobic adaptations. This potential intensity issue has been most recently addressed by Li et al. [58], whereby, it was suggested that the intensity of HIIT in hypoxia can be matched in normoxia by adjusting the relative peak power output and lactate threshold based on graded exercise testing [58]. Unfortunately, given the diverse methods researchers employed to report exercise intensity, we were unable to conduct a meta-regression on the effect of exercise intensity (and thus training load) on ∆VO_2max_, or a meta-regression on the effect of time in particular training zones on ∆VO_2max_. The exact mechanisms by which HIIT in hypoxia impacts VO_2max_ compared to normoxia remains controversial [59,60,61]. Increased transcription of skeletal muscle proteins involving redox regulation and glucose uptake has been reported [62], whilst training in hypoxia also improves mitochondrial function and subsequent ATP production [63]. All of these adaptations lead to the improved aerobic capacity shown with acute hypoxia training vs. normoxia training [59,63].

### 4.3. Impact of Altitude Extent and Intervention Duration on VO_2max_

While meta-regression analysis did not identify a relationship between altitude extent and SMD, an increase in VO_2max_ was observed following hypoxic HIIT vs. normoxic HIIT based on the SMD. A statistical explanation of the lack of dose (effective FiO_2_)-response (SMD) may be that variance was small between studies, with six of the nine studies utilising an effective FiO_2_ of 15.0–15.3%. Therefore, this limited range would unlikely explain the variance in SMD magnitudes between studies.

### 4.4. Limitations

A key limitation of this systematic review and meta-analysis was the lack of studies concerning HIIT in hypoxia. A greater number of studies would increase robustness of results and add weight to conclusions [64]. Thus, conclusions herein are preliminary until a greater body of literature surrounding HIIT in hypoxia and effect upon VO_2max_ is available. Moreover, authors sought to examine moderating effect of exercise intensity on the VO_2max_, however included studies displayed large variability in exercise prescriptions. Therefore, it remains unknown if different training loads modify the effect of hypoxia on VO_2max_. Although a limitation to this review, it is first and foremost a limitation within the field of study, as there are countless methods of measuring exercise intensity. Across studies, there were five different descriptions of intensity; 40% of the studies described intensity as % of maximal heart rate (HR_max_), 20% as rating of perceived exertion (RPE), 20% as % of lactate threshold (LAT), 10% as a % of maximum repetition (RM) and 10% as a % of velocity VO_2max_ (vVO_2max_). Similarly, repetitions, sets, and rest periods were seldom reported. It would improve future research if studies utilised the consensus on exercise reporting template (CERT [65]). Penultimately, as mentioned previously, six of the nine studies utilised an effective FiO_2_ of 15.0–15.3%. Moreover, only one of the ten studies examined an effective F_i_O_2_ of <14% (Ref. [48] F_i_O_2_ of 12.6%). Therefore, most studies considered moderate hypoxia, and the lack of variety in effective F_i_O_2_ in the included studies may be considered a limitation and an area for further research. Finally, study quality was low (PEDro scale mean ~4), and therefore this must be considered a limitation to the literature base. However, considering three points are awarded for blinding, it is difficult to blind participants to environmental altitude as conscious travel is required.

## 5. Conclusions

Findings from the present systematic review, meta-analysis, and meta-regression suggest that participating in HIIT in hypoxia improves VO_2max_ more than HIIT in normoxia. While there is a lack of association between effective FiO_2_ utilised and VO_2max_ improvement in the hypoxic groups, we believe this was a result of the limit range of FiO_2_ examined and therefore a statistical artefact. Thus, we believe it is pertinent to note that results from all but one study demonstrated a positive SMD (i.e., favouring HIIT in hypoxia vs. HIIT in normoxia).

## Figures and Tables

**Figure 1 ijerph-19-14261-f001:**
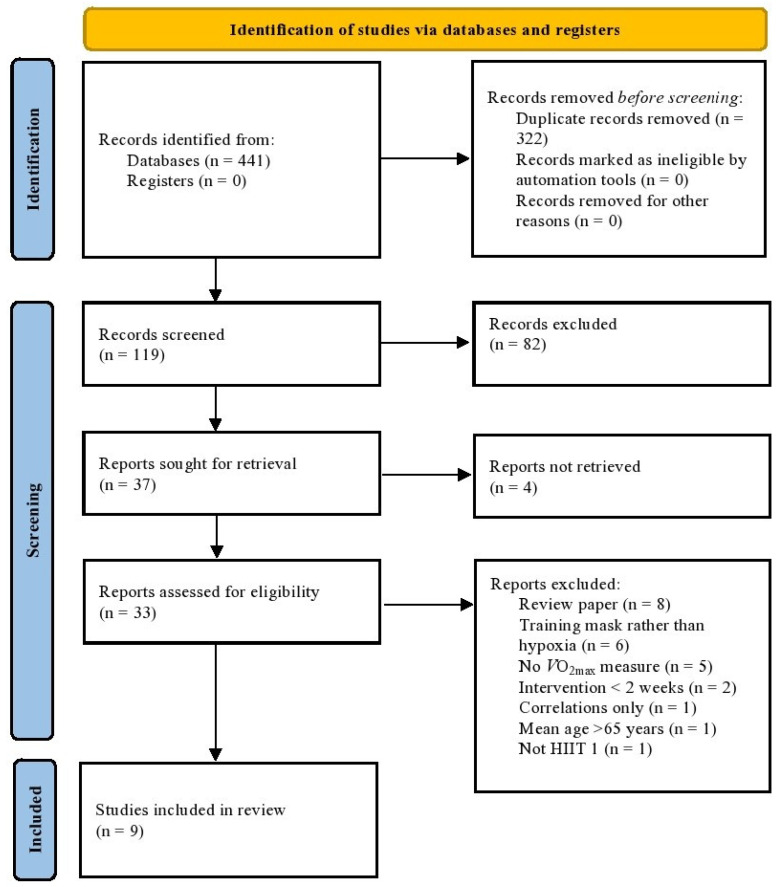
Preferred Reporting Items for Systematic Reviews and Meta-Analyses (PRISMA) flow diagram detailing inclusion and exclusion of potential studies as well as final number of studies included in the systematic review and meta-analysis. RCT = randomised control trial; CT = control trial.

**Figure 2 ijerph-19-14261-f002:**
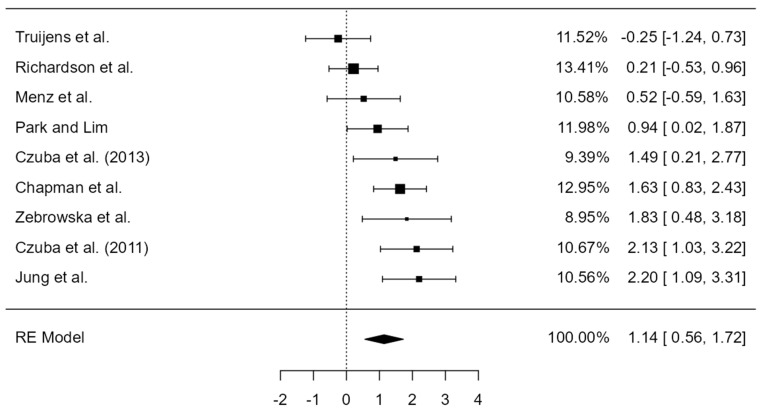
Summary of studies examining HIIT in hypoxia on VO_2max_ sorted by standardised mean difference (SMD). Data are presented as the percentage weight each study contributes to the pooled SMD, individual SMD [95% CIs]. Note that symbol size of individual studies is representative of the weighting for the pooled standardised mean difference. The filled diamond indicates overall SMD. RE = random effects [44,45,46,47,48,49,50,51,52].

**Figure 3 ijerph-19-14261-f003:**
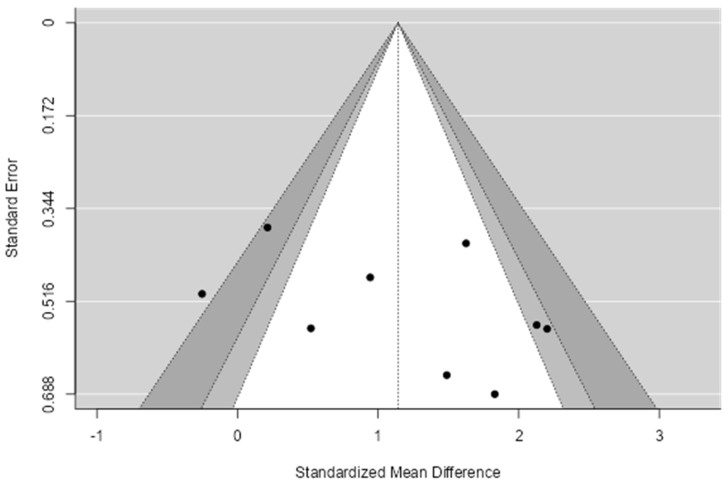
Funnel plot for evaluating publication bias in included studies examining HIIT in hypoxia and the effect on VO_2max_.

**Table 1 ijerph-19-14261-t001:** Description of included data sets, including exercise intervention protocol, study design, and VO_2max_ analysis method.

Study (Year)	Exercise; Exercise Intervention (Group)	Design Method	Intervention Duration (Weeks)	VO_2max_ Analysis Method	Study *n* (M/F)	Age (Years)	PEDro Scale
Chapman et al. [44]	Running; Interval training specific to athlete’s personalised mesocycle, intensity increased weekly up to 95% HR_max_ (Normoxic) Running; Interval training specific to athlete’s personalised mesocycle, intensity increased weekly up to 95% HR_max_ (Hypoxic, 2500–3000 m)	CT	13 Weeks	Douglas Bag; Modified Astrand-Saltin protocol	39 (27/12)	Total = 22 ± 3	4
Czuba et al. [45]	Cycling; HIIT: 15 min warm up @ 80% lactate threshold; 30–40 min @ 100% lactate threshold; 15 min active recovery @ 55% lactate threshold 120 min @ 60–75% lactate threshold, 3 d∙wk^−1^ for 3 weeks (Control) Cycling; HIIT: 15 min warm up @ 80% lactate threshold; 30- 40 min @ 95% lactate threshold; 15 min active recovery @ 55% lactate threshold; 120 min @ 60–75% lactate threshold, 3 d∙wk^−1^ for 3 weeks (Hypoxic; FiO_2_ = 15.2%, 2500–2600 m)	RCT	3 Weeks	Douglas Bag; Maximal incremental cycle ergometer protocol	20 (20/0)	Cont = 24 ± 4 Hyp = 22 ± 3	5
Czuba et al. [46]	Basketball; HIIT: 4–5 min, 90% vVO_2max_, 4 min active recovery, 60% vVO_2max,_ 3 d∙wk^−1^ for 4 weeks (Control) Basketball; HIIT:4–5 min, 90% vVO_2max_, 4 min active recovery, 60% vVO_2max,_ 3 d∙wk^−1^ for 4 weeks (Hypoxic, 2500 m)	RCT	3 Weeks	Breath by breath analysis: Maximal incremental cycle ergometer protocol	12 (12/0)	Cont = 22 ± 2 Hyp = 22 ± 2	5
Jung et al. [47]	Running; Interval training 90 min; 10 × 5 min, 90–95% HR_max_, 3 d∙wk^−1^ for 6 weeks (Normoxic) Running; Interval training 90 min; 10 × 5 min, 90–95% HR_max_, 3 d∙wk^−1^ for 6 weeks (Hypoxic, 3000 m)	RCT	6 Weeks	Breath-by-breath analysis; Bruce protocol	20 (20/0)	Norm = 26 ± 1 Hyp = 26 ± 2	4
Menz et al. [48]	One-Legged Cycling; HIIT: 4 × 4 min, 90% HR_max_, 3 d∙wk^−1^ for 3 weeks (Normoxic) One-Legged Cycling; HIIT: 4 × 4 min, 90% HR_max_, 3 d∙wk^−1^ for 3 weeks (Hypoxic, 4500 m)	RCT	3 weeks	Breath-by-breath analysis; Bruce protocol	13 (5/8)	Total = 26 ± 3	5
Park and Lim [49]	Swimming; Multi Component training 120 min: Running; 30 min 80% HR_max_, Cycling; 2 min × 10 90% HR_max_, Resistance training; multi set, multi exercise 80–90% 1 RM, 3 d∙wk^−1^ for 6 weeks (Normoxic) Swimming; Multi Component training 120 min: Running; 30 min 80% HR_max_, Cycling; 2 min × 10 90% HR_max_, Resistance training; multi set, multi exercise 80–90% 1 RM, 3 d∙wk^−1^ for 6 weeks (Hypoxic)	CT	8 Weeks	Breath-by-breath analysis: Maximal incremental cycle ergometer protocol	20 (10/10)	Norm = 23 ± 4 Hyp = 23 ± 3	4
Richardson et al. [50]	Cycling; Sprint Interval Training: 4–7 30 s, max effort, 3 d∙wk^−1^ for 2 weeks (Control) Cycling; Sprint Interval Training: 4–7 30 s, max effort, 3 d∙wk^−1^ for 2 weeks (Normoxic) Cycling; Sprint Interval Training: 4–7 30 s, max effort, 3 d∙wk^−1^ for 2 weeks (Hypoxic)	RCT	2 Weeks	Breath by breath analysis: Maximal incremental cycle ergometer protocol	42 (27/15)	Cont = 20 ± 1 Norm = 20 ± 1 Hyp = 20 ± 1	5
Truijens et al. [51]	Swimming; front crawl HIIT in flume, 10 × 30 s max effort RPE, 5 × 1 min, 5 × 30 s max effort, 6 d∙wk^−1^ for 5 weeks (Normoxic gas mix) Swimming; front crawl HIIT in flume, 10 × 30 s max effort, 5 × 1 min, 5 × 30 s max effort, 6 d∙wk^−1^ for 5 weeks (Hypoxic gas mix)	RCT	5 Weeks	Breath-by-breath analysis: Maximal incremental cycle ergometer protocol	16 (6/10)	Norm = 29 ± 9 Hyp = 29 ± 12	6
Zebrowska et al. [52]	Cycling; HIIT: 5 min warm up @30 W; 6 × 5 min @ 120% lactate threshold; 5 min intermittent recovery, 3 d∙wk^−1^ for 3 weeks (Normoxic) Cycling; HIIT: 5 min warm up @30 W; 6 × 5 min @ 120% lactate threshold; 5 min intermittent recovery, 3 d∙wk^−1^ for 3 weeks (Hypoxic)	RCT	6 Weeks	Brath by breath analysis: Maximal incremental cycle ergometer protocol	12 (12/0)	Total = 24 ± 4	5

RCT, randomised control trial; CT, control trial; HR_max_, Maximum Heart Rate; vVO_2max_, Velocity at VO_2max;_ s, seconds; Hyp, hypoxic; Cont, control; Norm, normoxic; HIIT, high intensity interval training; RPE, rate of perceived exertion.

**Table 2 ijerph-19-14261-t002:** Summary of hypoxic training conditions, VO_2max_ pre/post intervention, % ∆VO_2max_, altitude condition, administration, and total hypoxic dose.

Study (Year)	Altitude Conditions: FiO_2_ %	Hypobaric or Normobaric	Altitude Administration	Total Hypoxic Dose	Pre VO_2max_	Post VO_2max_	% Change
Chapman et al. [44]	Normoxic: FiO_2_ = 20.6 % Hypoxic: FiO_2_ = 14.3 %	Hypobaric		Natural altitude (2500 m) for 4 weeks.	Normoxic: 64.1 ± 4.4 mL·kg·min^−1^ Hypoxic: 65.0 ± 5.8 mL·kg·min^−1^	Normoxic: 64.4 ± 4.7 mL·kg·min^−1^ Hypoxic: 69.2 ± 6.8 mL·kg·min^−1^	+0.46% +6.46%
Czuba et al. [45]	Normoxic: FiO_2_ = 21% Hypoxic: FiO_2_ = 15.2%	Normobaric	Hypoxia Chamber	Simulated altitude (2500–2600 m) 3 weeks 3 × per week 60–80 min	Normoxic: 67.7 ± 2.0 mL·kg·min^−1^ Hypoxic: 67.8 ± 2.5 mL·kg·min^−1^	Normoxic: 67.5 ± 1.8 mL·kg·min^−1^ Hypoxic: 70.5 ± 1.5 mL·kg·min^−1^	−0.30% +3.98%
Czuba et al. [46]	Normoxic: FiO_2_ = 21% Hypoxic: FiO_2_ = 15%	Normobaric	Hypoxia Chamber	Simulated altitude (2500 m) 3 weeks 3 × per week 60 min	Normoxic: 42.2 ± 8.6 mL·kg·min^−1^ Hypoxic: 43.6 ± 7.9 mL·kg·min^−1^	Normoxic: 46.0 ± 7.5 mL·kg·min^−1^ Hypoxic: 48.8 ± 9.2 mL·kg·min^−1^	+9.01% +11.92%
Jung et al. [47]	Normoxic: FiO_2_ = 20.8% Hypoxic: FiO_2_ = 14.3%	Hypobaric		Natural altitude	Normoxic: 65.0 ± 4.1 mL·kg·min^−1^ Hypoxic: 63.2 ± 2.5 mL·kg·min^−1^	Normoxic: 66.1± 2.2 mL·kg·min^−1^ Hypoxic: 67.2 ± 3.2 mL·kg·min^−1^	+1.69% +6.33%
Menz et al. [48]	Normoxic: FiO_2_ = 21% Hypoxic: FiO_2_ = 12.6%	Normobaric	Hypoxia Chamber	Simulated altitude (4500 m) 3 weeks 3 × per week 30 min	Normoxic: 48.1 ± 12.4 mL·kg·min^−1^ Hypoxic: 45.4 ± 10.1 mL·kg·min^−1^	Normoxic: 50.1 ± 9.3 mL·kg·min^−1^ Hypoxic: 50.0 ± 9.8 mL·kg·min^−1^	+4.16% +10.13%
Park and Lim [49]	Normoxic: FiO_2_ = 20.6% Hypoxic: FiO_2_ = 14.3%	Hypobaric	-	Natural altitude (3000 m) 6 weeks 3 × per week 120 min	Normoxic: 58.1 ± 8.6 mL·kg·min^−1^ Hypoxic: 54.6 ± 6.6 mL·kg·min^−1^	Normoxic: 60.5 ± 7.2 mL·kg·min^−1^ Hypoxic: 60.1 ± 7.2 mL·kg·min^−1^	+4.13% +10.07%
Richardson et al. [50]	Normoxic: FiO_2_ = 21% Hypoxic: FiO_2_ = 15%	Normobaric	Hypoxia Chamber	Simulated altitude (2500 m) 2 weeks 3 × per week 22–35 min	Normoxic: 42.2 ± 8.6 mL·kg·min^−1^ Hypoxic: 43.6 ± 7.9 mL·kg·min^−1^	Normoxic: 46.0 ± 7.5 mL·kg·min^−1^ Hypoxic: 48.8 ± 9.2 mL·kg·min^−1^	+9.01% +11.92%
Truijens et al. [51]	Normoxic: FiO_2_ = 20.9 % Hypoxic: FiO_2_ = 15.3%	Normobaric	Inspiratory mouthpiece	Simulated altitude (2500 m) 5 weeks 3 × per week 30 min	Normoxic: 3.05 ± 0.58 l·min^−1^ Hypoxic: 2.92 ± 0.57 l·min^−1^	Normoxic: 3.22 ± 0.48 l·min^−1^ Hypoxic: 3.03 ± 0.53 l·min^−1^	+5.57% +3.77%
Zebrowska et al. [52]	Normoxic: FiO_2_ = 20.6% Hypoxic: FiO_2_ = 15.2%	Normobaric	Hypoxia Chamber	Simulated altitude (2500 m) 3 weeks 3 × per week 80 min	Normoxic: 54.2 ± 2.6 mL·kg·min^−1^ Hypoxic: 53.2 ± 2.6 mL·kg·min^−1^	Normoxic: 55.4 ± 4.8 mL·kg·min^−1^ Hypoxic: 59.4 ± 4.6 mL·kg·min^−1^	+2.21% +11.65%

## Data Availability

Publicly archived datasets analyzed or generated can be found here: https://studentmailuwsac-my.sharepoint.com/:u:/g/personal/lawrence_hayes_uws_ac_uk/EStBh1N-t9lBkpSc2PA-eDgBpzw-lVinRf454R5XEemgjg?e=JfhiWt (accessed on 2 January 2022).

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
