# Peer review of "High-Intensity Interval Training (HIIT) in Hypoxia Improves Maximal Aerobic Capacity More Than HIIT in Normoxia: A Systematic Review, Meta-Analysis, and Meta-Regression"

_ijerph, 2022, doi:10.3390/ijerph192114261_

Round 1

Reviewer 1 Report

This manuscripts evidences a lot of careful thoughts to conceptualize and deliver the output. There are only very few comments that I have and if addressed, will enhance the completeness of the piece. 

ABSTRACT:

1. specify total number of participants

2. Make a statement regarding the importance/application potential for your findings

INTRODUCTION: Page 2 Line 45: need to expand on how/whether this 1-3% increase is a significant improvement in terms of performance differences

RESULTS: you accounted for/looked into hypoxia duration effects. How about any other covariate analyses? e.g. accounting for gender split differences or age effects? 

DISCUSSION: Page 11 line 269: please rephrase to 'the low number of studies included'

Reviewer 2 Report

The authors of this systematic review and meta-analysis aimed to determine the effect of HIIT in hypoxic conditions vs HIIT in normoxia on VO2max. The review is well-written and conducted but a few things must be addressed before being endorsed for publication. This includes updating the search strategy and rerunning the search and undertaking sensitivity analyses to determine the robustness of findings.

Line 53, include a reference for the definition of HIIT. SIT is a derivative of SIT but different in that HIIT is typically below 100% Vo2max PMID:33760255

Line 58, please include the confidence interval. 

Line 71 to 74. Please soften to say that “LLTH has been shown to….”

Line 75 to 77. Please change structure of sentence. It reads as if the pervasive belief is that HIIT improves VO2 and secondary, hypoxia augments this (null result). It could be written along the lines of: Despite previous reports suggesting training in hypoxia can augment HIIT-induced adaptations in VO2max (REF), Hopeller et al….…. “

Line 78, how many more studies did you include over Hamlin’s study?

Line 104, please confirm whether you included studies with direct or indirect assessments of VO2max.

Line 109 to 116. Search needs to be updated. Search strategy needs to be restructured. I suggest: (“HIIT” OR “high-intensity interval*” OR “Interval training” OR “Interval exercise” OR “Sprint interval training”) AND “altitude” OR “hypox*”

AND “hypoxia”, “high-intensity interval training” AND “hypoxic”, 113 “high-intensity interval training” AND “altitude”, “Sprint interval training” AND 114 “hypoxia”, “Sprint interval training” AND “hypoxic”, and “Sprint interval training” 115 AND “altitude”. 

111 AND “hypoxia”, “HIIT” AND “hypoxic”, “HIIT” AND “altitude”, hypoxia”,

Line 155 to 164. Please confirm the method used for determining SMD. Was it the SMD of the difference or change in pre and post VO2 hypoxia vs difference or change in pre and post VO2 normoxia

Section 3.3, please conduct a sensitivity analysis by removing the two studies outside the funnel plot (I believe Truijens and Richardson) to see whether results change? Need to rewrite the publication bias section also to include the two studies which fell outside of the plot. Also, please conduct another sensitivity analysis to see whether studies for which ∆VO2max was calculated by the reviewers affected the results (ie remove them from the analysis). 

Line 204. The regressions are not useful because there isn’t enough between-study variability (largely due to small amount of studies) to determine an effect. This is particularly pertinent to some of the conclusions made in the abstract which are stronger than the data would permit. Maybe by quantifying training load (converted to load or METs) instead of training duration you could find better variability. 

Line 222. Change “with” to “in which” or change tense of demonstrated. 

Line 225 to 229. As per my previous comment, here you make the suggestion that training load may be predictive of change in VO2max but you can verify this through meta-regression. 

Line 261 to 267: thanks for acknowledging this. Soften references to this particular analysis throughout manuscript. 

Line 271, check double reference [52] (Sandeloski, 1995).

Line 274: what is SDM? Do you mean SMD?… Could be written as the authors aimed to examine the moderating effects of exercise intensity on changes in cardiorespiratory fitness (or VO2max)… to this end, quantifying exercise load would give an indirect answer to this question as volume and intensity are functions of load. 

Nice conclusion. 

Regards,

AS

Round 2

Reviewer 2 Report

Dear authors,

Thank you for for your careful consideration of my comments and speed of response.

I think the regression isn't overly helpful but it is conducted appropriately so I will let you decide whether to include or exclude it. My concern with it is that lay people interested in hypoxic training will read those results and take them as gospel without having adequate understanding of the limitations. 

With regards to the search proposal I only intended to write the first line (“HIIT” OR “high-intensity interval*” OR “Interval training” OR “Interval exercise” OR “Sprint interval training”) AND “altitude” OR “hypox*”). Please disregard the rest. I ran the search in Embase and retrieved 238 hits. See attached. 

Please advise how you wish to proceed with the two above queries but I am largely satisfied with the changes. I will recommend minor edits for you to have another opportunity to change the above should you decide to, otherwise I am happy to endorse for publication. 

All the best. 

Author Response

Dear R2,

Thank you for revisiting our submission.

We agree that the regression is not very informative (because of the data entry). However, the information that the regression is not very informative is useful in itself (I hope I have not overcomplicated this sentence too much!). What we mean is, without the M-regression, the reader may think 'what about a dose-response?'. However, that we have presented the M-regression informs the reader that a dose-response is not investigable given the current state of the literature included. To address this more explicitly, in the abstract we have written: "Neither extent of hypoxia, nor training duration modified this effect, however the range in FiO2 was small, which limits interpretation of this meta-regression." Similarly, in the discussion, we have written "While there is a lack of association between effective FiO2 utilised and V̇O2max improvement in the hypoxic groups, we believe this was a result of the limit range of FiO2 examined and therefore a statistical artefact." We believe this conveys our interpretation of the M-regression.

Thank you for clarifying the second point. We have rerun the search on PubMed (160 hits) and SPORTDiscuss (44 hits), and added to your search of Embase (238), we arrive at 442 hits. Our initial search resulted in 441 but was over a year ago. Therefore, we believe that we would not have yielded extra hits by searching the combined text, rather than the searches individually phrased. However, we would re-run the searches and filtering if you believe it neccessary.

Many thanks again and best wishes,
Lawrence